# Social Return on Investment (SROI) Evaluation of Citizens Advice on Prescription: A Whole-Systems Approach to Mitigating Poverty and Improving Wellbeing

**DOI:** 10.3390/ijerph22020301

**Published:** 2025-02-17

**Authors:** Rachel Granger, Ned Hartfiel, Victory Ezeofor, Katharine Abba, Rhiannon Corcoran, Rachel Anderson de Cuevas, Benjamin Barr, Aregawi Gebremedhin Gebremariam, Roberta Piroddi, Clare Mahoney, Mark Gabbay, Rhiannon Tudor Edwards

**Affiliations:** 1Centre for Health Economics and Medicines Evaluation, School of Health Sciences, Bangor University, Bangor LL57 2PZ, UK; rachel.granger@bangor.ac.uk (R.G.); ned.hartfiel@bangor.ac.uk (N.H.); v.ezeofor@bangor.ac.uk (V.E.); 2Institute of Population Health, University of Liverpool, Liverpool L69 3BX, UK; kabba@liverpool.ac.uk (K.A.); rhiannon.corcoran@liverpool.ac.uk (R.C.); rma@liverpool.ac.uk (R.A.d.C.); b.barr@liverpool.ac.uk (B.B.); a.gebremariam@liverpool.ac.uk (A.G.G.); r.piroddi@liverpool.ac.uk (R.P.); mbg@liverpool.ac.uk (M.G.); 3NHS Cheshire and Merseyside Integrated Care Board, Warrington WA1 1QY, UK; clare.mahoney@cheshireandmerseyside.nhs.uk

**Keywords:** social return on investment (SROI), Citizens Advice, socio-economic deprivation, wellbeing, non-clinical services for health, signposting, advice on prescription

## Abstract

Citizens Advice on Prescription (CAP), a Liverpool (UK)-based service, provides welfare advice and link worker social prescription support to people experiencing and at risk of experiencing financial or social hardship. CAP, which receives referrals from healthcare and third-sector services, aims to improve service users’ financial security, health, and wellbeing. A mixed-methods social return on-investment (SROI) analysis was used to evaluate this service. Between May 2022 and November 2023, a subset of service users (*n* = 538) completed the Short Warwick–Edinburgh Mental Wellbeing Survey (SWEMWBS) at baseline and a 2-month follow-up. Supporting quantitative and qualitative economic data were also collected (February 2023–February 2024) through semi-structured interviews (*n* = 16). Changes in social value were determined by comparing pre- and post-SWEMWBS scores. These scores were then mapped to monetary values using the Mental Health Social Value Bank (MHSVB). SROI ratios were then calculated by dividing the change in social value by the associated service provision costs. The mean social value change per person ranged from GBP 505.70 to GBP 697.52, and the mean service provision cost was GBP 148.66 per person. The overall study reported a positive SROI return range of GBP 1: GBP 3.40–GBP 4.69. The results indicate that non-clinical support services, like CAP, may be an effective intervention for addressing the wider determinants of health and wellbeing.

## 1. Introduction

The association between socioeconomic deprivation and poor health and wellbeing is well established [1,2]. Reduced access to resources that promote or sustain health within economically deprived populations is linked to both increased mortality and long-term health issues (morbidity) throughout one’s lifespan [3,4]. Financial insecurity, due to irregular, reduced, or low income (including long-term unemployment), is linked to increased vulnerability to mental distress and common mental health disorders, including anxiety and depression [5,6]. The post-2007 global economic crisis and the UK political response of austerity measures, including cuts to public funding, as well as the reduction in wages in real terms, have had a prolonged negative socioeconomic impact [7]. Financial pressures on the most vulnerable in society have also been compounded by the major welfare changes implemented with the 2012 Welfare Reform Act and the introduction of the Universal Credit system [8,9]. The detrimental impact of these welfare reforms on chronic health conditions, including mental health and wellbeing, have been widely reported [10,11,12]. In more recent years, the socioeconomically vulnerable have been hit harder by both the health and economic impacts of the COVID-19 pandemic as well as the rise in fuel prices and food costs driven by a combination of global events [8]. In the 2022–2023 fiscal year, real household disposable income, a key indicator of living standards, declined by 2.5%. This combination of increased cost-of living pressures and reduced public funding, including reduced welfare support, is a driver of the expanding gap between the poorest and most affluent in UK society [9,10].

Socioeconomic disparity between the North and the South of the UK is also well documented [13], with current differences in life expectancy of up to 20 years for females and 27 years for males between affluent Southern areas and deprived Northern areas of the UK [14]. Lower levels of life expectancy are particularly clustered in urban areas in the North of England in major cities such as Liverpool, Leeds, Newcastle, Manchester, and Blackpool [15]. Liverpool has some of the highest overall levels of socioeconomic deprivation in the UK. Out of 317 English local authorities, Liverpool ranks third in terms of health deprivation and disability, fourth for income deprivation and poverty, and fifth for employment deprivation [16].

The root cause of the significant proportion of health and social care usage in socioeconomically deprived populations is not only poor health or wellbeing but also seeking advice on non-clinical issues, including inadequate housing, welfare support, and debt or legal issues [17]. People living in socioeconomic deprivation often approach General Practitioners (GPs) or other healthcare professionals for help with these problems, help that healthcare staff have neither the expertise nor time to provide [18,19].

In recent years, local authorities, the third sector, and academics have explored using non-clinical support services, such as welfare and legal services, and social prescription alongside healthcare services to improve individuals’ health and wellbeing. Since the Legal Aid, Sentencing and Punishment of Offenders (LASPO) Act of 2012, there has been a significant reduction in free welfare legal services in the UK, with charitable organisations, including Citizens Advice, becoming the principal providers [20]. Demand for citizens’ advice services has more than doubled in the last four years, with crisis support, including heating payments and access to food banks, reflecting significantly increased needs [21]. Social prescribing, also known as community referral, was developed as a non-medical intervention to provide support for an individual’s social, emotional, and practical needs [22]. Social prescription provision is currently part of the NHS contractual requirement for Primary Care Networks (PCNs), where healthcare professionals refer patients to a social-prescribing link worker, who helps identify a range of suitable non-clinical activities or support groups in the patient’s local community [23].

A range of small-scale studies have reported that welfare support interventions do improve financial security, with the monetary benefits being greater than the cost-of-service provision [24,25,26]. Many of these studies have reported improved mental health and reduced stress-related conditions for service users, although the quantitative impact of these mental-health- or wellbeing-related changes on the cost effectiveness of services has not been reported. Social prescribing interventions have also been linked to improvements in anxiety levels, wellbeing, and quality of life, with these changes reported to be cost-effective [27] and linked to reductions in healthcare use [22].

Citizens Advice on Prescription (CAP) is a novel intervention that combines both welfare and social prescription support. It was developed as a collaboration between Citizens Advice Liverpool and Liverpool NHS to help to alleviate poverty and address the underlying causes of poor health [28]. The service, initially trialled in GP surgeries (2014), has since been extended to all Liverpool primary care GP surgeries (2015) and secondary care services, including mental health services (2015), cancer services (2015), respiratory services (2018), and perinatal services (2021). CAP also enables organisations (including the third sector) that support people experiencing mental distress to refer to this service. Once referred, a patient is contacted within two working days of the referral, and a full assessment is carried out within two weeks.

During the assessment, a trained advisor gathers detailed information about the service user’s financial situation, living conditions, social circumstances, and health concerns. This information helps the advisor create a personalised support plan, which may include benefits advice, debt management assistance, housing support, employment guidance, and referrals to other essential services.

The full CAP evaluation was a multi-methods study led by the University of Liverpool which included impact evaluation using instrumental variable analysis and the qualitative investigation of service users’ and stakeholders’ experiences of the service [submitted; in preparation]. The social return on investment (SROI) evaluation, led by Bangor University, evaluated the impact of the intervention on service users’ wellbeing and determined the cost-effectiveness of the service.

SROI is a pragmatic type of social cost–benefit analysis (CBA) commonly used to evaluate complex interventions [29,30]. SROI allows a bottom-up stakeholder perspective and a mixed-methods approach to outcomes that are relevant to stakeholders by allocating financial proxies for outcomes that often do not have market value [31]. The social CBA approach is recommended in His Majesty’s (HM) Treasury Green Book (UK) [32], with the framework for SROI described in the Cabinet Office ‘A Guide to Social Return on Investment’ [33].

## 2. Materials and Methods

The SROI evaluation process, as outlined in the 2012 Cabinet Office Guide [33], typically involves six key stages: (1) identifying stakeholders, (2) developing a theory-of-change model, (3) calculating inputs, (4) evidencing and valuing outcomes, (5) determining the impact, and (6) calculating the SROI ratio. Each of these stages is discussed below.


**Stage 1: identifying stakeholders**


As the aim of the SROI evaluation was to evaluate the impact of CAP on service users’ wellbeing, service users were the only stakeholders included in this SROI evaluation. Eligible participants were patients (aged 16 or over) who were identified by health service professionals as being at financial risk or requiring social support and who were referred to CAP. Participants had the capacity to provide informed consent, were able to speak English, and were able to complete the Short Warwick–Edinburgh Mental Wellbeing Survey (SWEMWBS) questionnaire during phone or in-person interviews with CAP staff.

Due to the wide range of referral routes into the service, sub-analysis was also carried out using the main referral pathways. Primary care referrals were from GP surgeries and made either by GPs or other health professionals. Secondary care referrals were made through the various secondary care referral pathways as well as third-sector mental health organisations in the Liverpool area. Perinatal referrals came through health visitors, midwives, children’s centres, and perinatal mental health services. The perinatal service also worked closely with the Non-English-Speaking Team (NEST) at Liverpool Women’s Hospital to provide the service for this particularly vulnerable population.


**Stage 2: developing a theory-of-change model**


Theory-of-change models are a way to explore the linkages and underlying assumptions between inputs, outputs, and expected outcomes of an intervention [34]. The theory-of-change model shown in Figure 1 was developed for the CAP trial by the University of Liverpool research team (the findings from this trial are currently being published). The primary outcomes predicted based on the model were reduced mental health problems and improved overall wellbeing.


**Stage 3: calculating inputs**


Costs for CAP service provision were the only costs included in this evaluation. Service costs were calculated as the average service costs per service user, using total service costs divided by the total number of service users for the financial year 2022–2023 (the only full financial year during the evaluation).


**Stage 4: evidencing and valuing outcomes**


The outcome for the SROI was the improvement in service users’ mental wellbeing, measured using the SWEMWBS questionnaire. SWEMWBS measures a range of aspects that affect mental wellbeing, including levels of stress, social connection, and self-confidence [35]. The questionnaire consists of seven statements: *I’ve been feeling optimistic about the future; I’ve been feeling useful; I’ve been feeling relaxed: I’ve been dealing with problems well; I’ve been thinking clearly; I’ve been feeling closer to other people; I’ve been able to make up my own mind about things*. Participants are asked to answer based on how they have been feeling in the last two weeks, giving a rating for each question ranging from 1 (none of the time) to 5 (all the time), with a possible overall score range of 7 to 35.

To calculate social value using the SWEMWBS questionnaire, participants complete the questionnaire at baseline and follow-up time points. Total scores are calculated for both time points. Each total score is assigned a monetary value, and the change in value between baseline and follow-up is determined. Finally, 27% is subtracted from this change to account for deadweight (what would have happened anyway), resulting in the final social value for each participant.

Mental wellbeing valuation using the SWEMWBS questionnaire is an established and robust method for estimating the financial (social) value of health- and wellbeing-related outcomes that do not have a market value [32]. The mental health social value calculation is used to allocate financial valuations to the SWEMWBS total score, and changes in SWEMWBS scores at baseline and follow-up are used to calculate the change in social value generated. The monetary values below (see Table 1) indicate the amount of money the average person would need to receive in order to experience an equivalent increase in their overall SWEMWBS score [36].

SWEMWBS questionnaires were collected by CAP staff. Service users were asked if they consented to be part of the evaluation and complete a baseline SWEMWBS questionnaire in their initial CAP interview. They were asked if they agreed to be contacted after approximately 2 months to complete a follow-up questionnaire. Follow-up contact was attempted up to 3 times. SWEMWBS questionnaires were completed predominantly by phone, with face-to-face interviews used for some referrals. Only service users that completed both baseline and follow-up questionnaires were included in the SROI evaluation.


**Stage 5: determining impact**


When evaluating complex interventions and/or interventions in complex systems, it is important that steps are taken to reduce the risk of overstating the benefits of the intervention. In SROI evaluations, this is addressed by considering the following factors:

**Deadweight:** The acknowledgement that there is likely to be a proportion of the reported outcomes that would have happened anyway.

**Attribution:** The proportion of the outcome directly attributed to the intervention by participants rather than other factors in their lives.

**Displacement:** Whether participants had to give up any other activities that could have contributed to the outcomes in order to take part in the intervention.

SROI evaluations also consider the drop-off of the intervention, i.e., a measure of how long the impact of the intervention is expected to last. A drop-off measure of a 1-year time frame was estimated for this evaluation, based on previous studies, and validated by the additional health economic questions included in this study.

A 27% standard deadweight percentage was subtracted from the social value calculated from the change in SWEMWBS scores, as recommended in the established methodology [36]. Levels of attribution, displacement, and drop-off were measured through the inclusion of additional health economics questions, which are provided as Appendix A. These data were collected alongside data on the qualitative investigation of service users’ experiences of the CAP service. Reported levels of attribution and displacement were used to calculate a sensitivity range for the SROI evaluation. The drop-off measure was used to validate the 1-year drop-off range estimated for this SROI evaluation.


**Stage 6: calculating the SROI**


SROI ratios were calculated to compare the SWEMWBS-based social value changes per participant, compared to the service cost per participant, with the SROI ratio expressed as the social value created per GBP 1 invested in the service (see Equation (1)).SROI ratio = social value change per participant/cost of CAP service per participant(1)

Sensitivity analysis was also carried out to estimate the social value change estimated by participants to be due to the CAP service as opposed to any other factors (e.g., measures of attribution or displacement) (see Equation (2)).SROI ratio = social value change reported per participant − (% attribution + displacement)/cost of CAP service per participant(2)

## 3. Results

### 3.1. Demographics of Service Users and SWEMWBS Responses

SWEMWBS data were collected over a 20-month period (May 2022–November 2023). During this time, there were a total of 13,294 CAP service users, of which 2133 (16%) completed the SWEMWBS at baseline, and 538 (4%) completed the SWEMWBS at baseline and at the follow-up. Only the 538 respondents who completed both baseline and follow-up questionnaires were included in the SROI analysis, but total service user and baseline SWEMWBS respondent demographics are also provided as comparators (see Table 2).

Primary care referrals accounted for 84.0% of the total service users, with secondary care and perinatal referrals accounting for 7.5% and 8.5% of service users, respectively. Both baseline SWEMWBS respondents and SROI respondents had comparative levels of primary care referrals (around 82%) but were under-representative of secondary care referrals (around 5%) and over-representative of perinatal referrals (around 12%).

Females accounted for a higher proportion of total service users (1:1.6) in all pathways. This higher service usage may indicate that women are more likely to be eligible for the CAP service due to financial or social hardships. The proportion of women was also incrementally higher among baseline SWEMWBS respondents and SROI respondents (1:2.0 and 1:2.1, respectively), meaning women were over-represented in the SROI evaluation. The ratio of individuals of white/ethnic minority ethnicities was consistent (at around 1:0.2) across primary care and secondary care referrals, which is in line with the reported ethnicity distribution of the Liverpool population (of 84%:16% white/ethnic minority) [37].

With regard to perinatal referrals, ethnic minority individuals made up around half of all service users (with a ratio of 1:1.1), reflecting the fact that many service users were referred through the Non-English-Speaking Team (NEST) service. However, the proportions of ethnic minority participants were lower among the baseline SWEMWBS and SROI respondents (at 1:0.7 and 1:0.8, respectively), indicating that ethnic minority perinatal service users were under-represented in the SROI evaluation of perinatal services.

The average service user age was 51.7 (±17.0), with the age of primary care referrals (52.4 ± 17.1) being marginally higher than that for secondary care referrals (47.9 ± 10.1). Perinatal referrals were younger (32.3 ± 6.7), as expected for this cohort. The age of the baseline SWEMWBS respondents and SROI respondents was generally in line with that for the total service users.

The baseline SWEMWBS scores for the primary care route were slightly higher than those for secondary care referrals (18.1 and 17.1, respectively) and similar for SROI respondents (19.0 and 16.9, respectively), indicating that the baseline wellbeing of those that were included in the SROI evaluation was representative of the larger number of baseline SWEMWBS respondents. The baseline SWEMWBS scores for perinatal care were notably higher, at 24.7, and marginally higher, at 25.9, for SROI respondents. Only primary care and secondary care referrals reported a change from baseline to the 2-month follow-up (an increase of 0.9 and 1.5, respectively), with perinatal referrals reporting no change (0.0).

### 3.2. Total Service Costs and Average Service User Costs

The total CAP service cost for the financial year 2022–2023 was GBP 1,254,417, with the majority (89%) spent on covering staff costs (see Table 3). There was an overhead cost of GBP 126,646 (10%), which includes office costs as well as room hiring in GP surgeries (GBP 10,800) to provide CAP services in primary care locations. Also included were community partner payments (GBP 29,600), a nominal support payment made to the five social prescribing services most referred to by CAP link workers each month. Routine data administration and evaluation costs made up GBP 10,840 (1%) of service costs.

From April 2022 to March 2023, there were 8438 CAP service users. The average service cost per service user was calculated by dividing the total costs for the 2022/2023 financial year by the number of service users, yielding an average service cost per service user of GBP 148.66.

### 3.3. Calculating the Social Value Change and SROI Ratios

Due to the comparatively large number of service users included in this SROI evaluation (*n* = 528), the total social value changes, along with mean changes in social value, are reported in Table 4. Individuals’ social value changes are also available from the corresponding author on request.

A standard 27% was deducted from the sum of social value changes to account for deadweight (i.e., what would have happened anyway, regardless of the CAP intervention). This was then divided by the number of service users per pathway, yielding the average social value change per service user for each of the service pathways as well as the overall service.

There was a positive change in social value (amounting to GBP 697.52 per person) reported for the overall service, with secondary care referrals exhibiting the largest average social value change (GBP 1802.75), followed by primary care referrals (GBP 701.63). and perinatal referrals (GBP 133.73).

SROI ratios were then calculated by dividing the average social value change per person by the average service cost per person. The overall service yielded a positive SROI return of GBP 1: GBP 4.69 (i.e., GBP 4.69 of social value was created for every GBP 1 invested in the service). Both primary care and secondary care referrals yielded a social value gain for each GBP 1 spent on the service (GBP 1: GBP 4.72 and GBP 1: GBP 12.13), whereas for perinatal referrals, the social value gain was less than the cost-of-service provision (GBP 1: GBP 0.90).

### 3.4. Sensitivity Analysis: Consideration of Attribution and Displacement

Information on attribution, displacement, and drop-off was collected as part of the semi-structured 1-to-1 qualitative interviews (*n* = 9, out of 13 possible interviewees). An additional focus group was also established to collect information from perinatal referees (*n* = 7), as no perinatal referees had been recruited for 1-to-1 interviews.

Of these 16 service users, 14 reported the percentage of change in their wellbeing they attributed to the service (attribution), with participants across primary care and secondary care reporting, on average, a higher level of attribution (72.5%) compared to those using the perinatal referral pathway (60.0%). The level of attribution was used to perform the sensitivity analysis for the SROI.

All 16 respondents reported on displacement, i.e., the impact of the service on other factors that might impact their wellbeing. Only one person reported that they had had to give up other activities that may have affected their wellbeing to take part in the service; the level of this impact was reported to be small. This indicates that there was negligible displacement due to the service, so an adjustment for displacement did not need to be included as part of the sensitivity analysis.

Of the 14 service users who reported on the expected duration of the service impact (drop off), the majority (12) thought that the impact would last for 6–12 months or over 12 months, which is in line with the 1-year time horizon used for the SROI evaluation. All individual data on attribution, displacement, and drop-off are provided as Appendix A.

Given the limited sample size of interviewees who provided attribution data (*n* = 14), a sensitivity analysis was conducted utilising the reported attribution levels (72.5% for primary care and secondary care and 60% for perinatal) to establish a potential range of SROI values. Since the primary care and secondary care pathways represent over 90% of the referrals, an attribution level of 72.5% was used for the sensitivity analysis. This resulted in an overall SROI range of GBP 1: GBP 3.40–GBP 4.69.

### 3.5. Qualitative Feedback on the CAP Service

The interview participants also had the opportunity to give some qualitative feedback on the service as part of the additional health economic questions. The feedback was that the service was highly valued and convenient, and the participants intended to continue to use the service for further support. They felt that it put them in a better position to take up other opportunities, and some mentioned that they had not realised how stressed they were until the pressure was relieved. All but one service user who provided qualitative feedback (*n* = 14) reported feeling that their wellbeing had improved since using the service, with the improvement mainly stemming from using the service.

However, the participants also highlighted ongoing concern about the impact of future financial stressors, including Personal Independence Payment (PIP) reviews, redundancy risks, and further financial strain over school holidays, indicating that ongoing access to CAP is likely to be an important aspect of the service for some service users.

## 4. Discussion

### 4.1. Overall Findings

The findings of this study reveal that the CAP service, using a combination of welfare advice and link-worker social prescription support, does improve service users’ wellbeing as measured via the SWEMWBS questionnaire, with an overall positive SROI of GBP 1: GBP 3.40–GBP 4.69 reported. A novel finding of this study is that there is a notable difference in SROI according to service referral pathway. Although secondary care referrals yielded the highest SROI (GBP 1: GBP 8.79–GBP 12.13), the number of respondents was low (*n* = 31), and they were referred by a disparate range of secondary referral services, with around half coming from mental health services, while referrals from cancer and respiratory services were also noted. Because of this, further work is required to understand the potential of this service through the various secondary care services currently referring individuals to CAP.

The level of SROI reported by primary care referrals (GBP 1: GBP 3.42–GBP 4.72) is particularly important, as 84% of CAP service users are referred through this pathway. Although there is an over-representation of female respondents for all the SROI-included referrals, the demographics of the SROI-included primary care referrals are generally in line with total service primary care referrals, indicating that the included respondents are likely to be representative of total service users and this level of return is likely to be a realistic measure of total CAP service referrals through this pathway.

In comparison, perinatal referrals reported SROI ratios of less than 1, as the social value gain was less than the service costs (GBP 1: GBP 0.54–GBP 0.90). Perinatal referrals differed from primary care and secondary care in having comparatively high SWEMWBS scores at baseline and reporting no change at the follow-up. There could be various explanations as to why we see very little change in wellbeing in the perinatal group. A decline in wellbeing in the absence of the intervention might be expected since these service users have generally experienced adverse social welfare events just prior to receiving support. Also, other social conditions were declining for families during this period due to the cost-of-living crisis. These service users were also undergoing the experience of having a new child, and this is likely to have influenced their wellbeing over time, particularly if post-natal depression or family disharmony arose.

One factor highlighted by the CAP team was the disproportionate level of inadequate-housing issues experienced by perinatal referrals (29%) compared to all CAP service users (12%) (corresponding data are from 2022–2023 CAP figures and personal correspondence). Qualitative research also highlighted that housing issues were experienced both by perinatal referrals and patients with young families referred though the other pathways (corresponding data are being published). These were often long-term issues, with patients reporting that although the CAP service was invaluable in providing support and driving improvements in living conditions, these changes and any linked improvements in wellbeing took considerably more time to arise than the two month follow-up period used in this study. So, if this group’s wellbeing were to have declined in the absence of the intervention, the maintenance of wellbeing reported in this study as a result of the intervention would be a positive outcome. However, more data covering a longer period are needed from this group to verify the impact of the intervention.

### 4.2. Comparison with Previous Studies

The SROI findings reported here are in line with previous UK studies on welfare advice and social prescribing services. Three previous SROI studies on Citizens Advice or similar welfare services reported ranges of GBP 1: GBP 3.31, GBP 1: GBP 6.01, and GBP 1: GBP 6.23 [27,38,39]. Evaluations of social prescribing have been reported to be cost-effective [27]. A community hub for chronic conditions in North Wales reported a positive SROI return of GBP 1: GBP 2.60–GBP 5.15 [40]. A small study of a social prescribing scheme receiving referrals from GP practices in London also reported a wellbeing improvement with an associated positive SROI return of GBP 1: GBP 3.51–GBP 8.56 [41]. So, the findings from this evaluation corroborate the evidence from previous smaller studies showing that welfare and social prescribing interventions may be effective in generating positive social value by improving service users’ mental health and wellbeing.

### 4.3. Strengths of the Study

This study employed a natural experiment approach. This means the CAP intervention investigated was not deliberately designed or controlled by the researchers. Instead, it was carried out naturally within a complex, real-world setting. Economic evaluations conducted alongside natural experiments are becoming a more recognised and needed approach to addressing the socioeconomic determinants of health and the evaluation of services to mitigate the adverse effects of socioeconomic hardship [31,42]. This study investigated a novel non-clinical intervention to improve wellbeing in socioeconomically disadvantaged populations, and we report that this approach is an effective and cost-effective intervention, with the change in wellbeing and the associated social value change being greater than costs for the service. We also highlight that the application of this non-clinical intervention showed significant variation across referral pathways, which is an important consideration for future service focus and potential future rollout of these types of services. As our evaluation included a relatively large sample for the SROI evaluation and the SROI participants were demographically representative of all service users, it seems likely that the findings are reliable. In addition, as the SROI was part of a multi-evaluation approach, we were able to represent data in the various ways necessary for a complex intervention, giving a more nuanced understanding of its impact.

### 4.4. Weaknesses of This Study

Although the overall sample size was relatively large for an SROI evaluation, we recognise that the sample analysed only represented 4% of the total service users in this period, with secondary care referrals under-represented and perinatal referrals overrepresented. The disparate nature of secondary services meant that it was not possible to determine if the positive return reported for this pathway would be comparable for all secondary services. We also acknowledge that the 2-month follow-up period used in this study was too short to capture many of the long-term issues and associated wellbeing changes that are experienced by the perinatal referrals.

The major limitations of this study were that mental wellbeing was assessed using only one outcome measure (SWEMWBS) and that the intervention took place in a complex environment with no control group. However, to account for the absence of a control group, a 27% deadweight percentage was applied. This percentage was derived from rigorous wellbeing valuation methodologies [43]. Nevertheless, the SROI methodology interprets change from baseline as a causal effect of the program, assuming that in the absence of the intervention, there would have been no change in the SWEMWBS score. This is probably an unrealistic assumption. People are referred to the CAP service because something has gone wrong; this often implies quite severe social welfare issues. It is implausible that wellbeing would have been unchanged following such issues in the absence of the intervention. So, no change in wellbeing or even a small decline in wellbeing could still be consistent with a large program effect on wellbeing.

## 5. Conclusions

The findings suggest that CAP, a welfare and social prescription support service, may be effective in generating positive social value by improving self-reported wellbeing among individuals experiencing or at risk of experiencing financial or social hardship. At a time of increasing socioeconomic disparity in the UK, analysts need to widen their evaluative spaces using natural experiments such as the one reported here.

## Figures and Tables

**Figure 1 ijerph-22-00301-f001:**
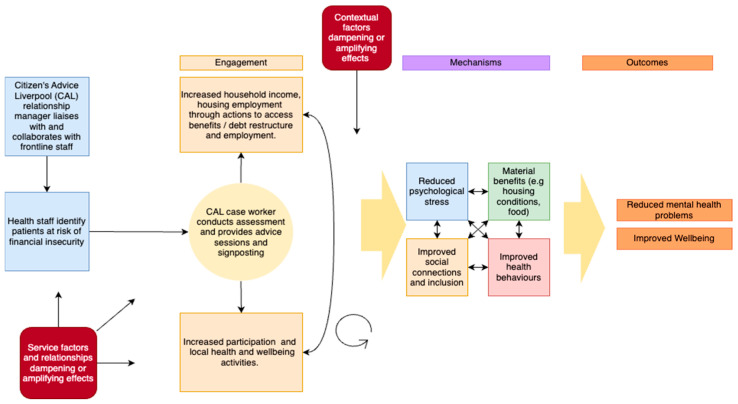
Theory-of-change model for the CAP intervention.

**Table 1 ijerph-22-00301-t001:** SWEMWBS scores and monetary values based on wellbeing valuation.

Overall SWEMWBS Score	Monetary Value Based on Wellbeing Valuation
7–14	GBP 0
15–16	GBP 9639
17–18	GBP 12,255
19–20	GBP 17,561
21–22	GBP 21,049
23–24	GBP 22,944
25–26	GBP 24,225
27–28	GBP 24,877
29–30	GBP 25,480
31–32	GBP 25,856
33–34	GBP 26,175
35	GBP 26,793

**Table 2 ijerph-22-00301-t002:** CAP service user demographics and SWEMWBS responses.

	Primary Care Referrals	Secondary Care Referrals	Perinatal Referrals	Total Referrals
**Total CAP service users**				
Number of service users	11,167	997	1130	13,294
% of service users	84.0%	7.5%	8.5%	
Gender: ratio of males/females *	1:1.4	1:1.1	1:17.3	1:1.6
Ethnicity: ratio of white/ethnic minority **	1:0.2	1:0.2	1:1.1	1:0.3
Age (±SD)	52.4 (±17.1)	47.9 (±10.1)	32.3 (±6.7)	51.7 (±17.0)
**Baseline SWEMWBS respondents**				
Number of respondents	1967	121	304	2393
% of baseline respondents	82.2%	5.1%	12.7%	
Gender: ratio of males/females *	1:1.6	1:1.5	1:22.3	1:2.0
Ethnicity: ratio of white/ethnic minority **	1:0.2	1:0.1	1:0.7	1:0.3
Age (±SD)	51.9 (±15.8)	52.8 (±12.4)	31.6 (± 6.6)	51.0 (±15.9)
Baseline SWEMWBS score (±SD)	18.1 (±6.5)	17.1 (±5.5)	24.7 (±6.8)	19.5 (±6.8)
**SROI included respondents**				
Number of respondents	443	31	64	538
% of SROI respondents	82.3%	5.8%	11.9%	
Gender: ratio of males/females *	1:1.8	1:0.9	1:31.0	1:2.1
Ethnicity: ratio of white/ethnic minority **	1:0.2	1:0.2	1:0.8	1:0.2
Age (±SD)	53.3 (±15.7)	49.2 (±12.9)	32.4 (±8.4)	52.3 (±16.0)
Baseline SWEMEBS scores (±SD)	19.0 (±6.4)	16.9 (±5.6)	25.9 (±6.4)	19.7 (±6.8)
Follow-up SWEMWBS scores (±SD)	19.9 (±6.7)	18.4 (±6.4)	25.9 (±6.3)	20.6 (±6.9)
Change in SWEMWBS scores (±SD)	0.9 (±5.9)	1.5 (±6.8)	0.0 (±5.8)	0.8 (±5.9)

Results are shown for total CAP service and for the three main referral pathways. * Gender is presented as a ratio of males/females to account for the missing gender classification data for total service users (5%) and SWEMWBS respondents (1%). ** Ethnicity is presented as a ratio of white/ethnic monitory individuals to account for the missing ethnicity classification data for total service users (9%) and SWEMWBS respondents (2%).

**Table 3 ijerph-22-00301-t003:** Costs for CAP service and average cost per service user.

Cost Type	Costs for Financial Year 2022–2023
Staff costs, GBP (%)	GBP 1,116,931 (89%)
Overhead costs, GBP (%)	GBP 126,646 (10%)
Service data collection and evaluation costs, GBP (%)	GBP 10,840 (1%)
Total Costs	GBP 1,254,417
Total unique service users, April 22–March 23	8438
Average service cost per service user	GBP 148.66

**Table 4 ijerph-22-00301-t004:** Changes in social value and SROI ratios for the CAP service.

Referral Pathway	Service Users (*n*)	Sum SV at Baseline	Sum SV at Follow Up	Change in SV	Change in SV—Deadweight (27%)	Average SV Change pp	Average Service Cost pp	SROI Ratio
Primary care referrals	443	GBP 6,144,876	GBP 6,570,657	GBP 425,781	GBP 310,820	GBP 701.63	GBP 148.66	GBP 4.72
Secondary care referrals	31	337,448	414,003	GBP 76,555	GBP 55,885	GBP 1802.75	GBP 148.66	GBP 12.13
Perinatal referrals	64	GBP 1,378,276	GBP 1,390,000	GBP 11,724	GBP 8559	GBP 133.73	GBP 148.66	GBP 0.90
Total service	538	GBP 7,860,600	GBP 8,374,660	GBP 514,060	GBP 375,264	GBP 697.52	GBP 148.66	GBP 4.69

SV = social value; pp = per person.

## Data Availability

All data have been made available within the paper or in the Appendix A.

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
