# Peer review of "Social Return on Investment (SROI) Evaluation of Citizens Advice on Prescription: A Whole-Systems Approach to Mitigating Poverty and Improving Wellbeing"

_ijerph, 2025, doi:10.3390/ijerph22020301_

Round 1

Reviewer 1 Report

Comments and Suggestions for Authors

Dear authors. Many thanks for this very interesting paper. I have some comments, especially asking for clarification to ease the paper's readiness:

Abstract

Please specify what the return is on. You mentioned that you used mental health scores, but can you clarify precisely in which outcomes the gains were observed? The abstract is full of acronyms. I would avoid that and introduce the complete names. 

Introduction

·      Please cite recent sources showing the budget cuts and decrease in real wages in the UK during the study period. It would be important to know how significant the cuts were and how much real wages have been reduced.

·      Can you clarify how the CAP assessment is done? You mention that once they get the referral, the assessment is done within two days. What information do they collect, how do they get the information… and what do they do for the assessment?

·      In the methodology, you subtract 27% of deadweight, assuming that would have been the change if nothing else happened. However, isn’t it possible to have data from a group not referred to and see how much outcomes have changed over time? Alternatively, many people may have been referred and never used the service. Is this a possibility?

·      Please, to facilitate readiness write down the formula of SROI and explain each component in order. 

Methods

·      Please explain in more detail what is collected in the SWEMWBS. Which items do you use and the scales, mostly for the outcomes used?

·      The authors must explain how the social value estimates were calculated. Please include the methodology in the paper. 

·      It is necessary to include some characteristics of the sample evaluated besides gender. It would be important to know how bad their health condition was, what condition they have, what is their family situation, what were their financial status, and their working conditions. This is relevant to understand who is really benefiting from the service. It may be that those who are better off take over the service, and those in worse situations avoid the service as they do not have time. 

Discussion

·      Could you please clarify why you consider this a natural experiment? People are referred to the CAP (treatment), but they can decide whether to get it or not. Also, referrals are made to people under certain circumstances. A natural experiment would randomly assign people to CAP without any anticipation, so it is possible to observe changes over time.

Author Response

Please see attached Word document

Reviewer’s comments

Response and action

Page number

Reviewer 1

Abstract

Please specify what the return is on. You mentioned that you used mental health scores, but can you clarify precisely in which outcomes the gains were observed? The abstract is full of acronyms. I would avoid that and introduce the complete names. 

We have reduced the number of acronyms in the abstract. 

We have edited the text to provide more clarity that the outcomes were observed by comparing pre- and post-SWEMWBS scores.

Changes in social value were determined by comparing pre- and post-SWEMWBS scores. These scores were then mapped to monetary values using the Mental Health Social Value Bank (MHSVB). Social Return on Investment (SROI) ratios were then calculated by dividing the change in social value by the associated service provision costs. The mean social value change per person ranged from £505.70 to £697.52 and the mean service provision cost was £148.66 per person. The overall study reported a positive SROI return ranging from £1: £3.40 - £4.69. Results indicate that non-clinical support services, like CAP, may be an effective intervention to address the wider determinants of health and wellbeing. 

Page 1

Introduction

·      Please cite recent sources showing the budget cuts and decrease in real wages in the UK during the study period. It would be important to know how significant the cuts were and how much real wages have been reduced.

We have added text to indicate the decrease in real household disposable income during the study period:

In the 2022-23 fiscal year, real household disposable income, a key indicator of living standards, declined by 2.5%.

Page 2

·      Can you clarify how the CAP assessment is done? You mention that once they get the referral, the assessment is done within two days. What information do they collect, how do they get the information… and what do they do for the assessment?

We have added the following text:

During the assessment, a trained advisor gathers detailed information about the service user’s financial situation, living conditions, social circumstances, and health concerns. This information helps the advisor create a personalised support plan, which may include benefits advice, debt management assistance, housing support, employment guidance, and referrals to other essential services.

Page 3

·      In the methodology, you subtract 27% of deadweight, assuming that would have been the change if nothing else happened. However, isn’t it possible to have data from a group not referred to and see how much outcomes have changed over time? Alternatively, many people may have been referred and never used the service. Is this a possibility?

This study's scope precluded the collection of pre- and post-SWEMWBS scores from individuals referred to CAP but who did not utilise its services. Pre- and post-SWEMWBS survey completion was unlikely among this group.

We acknowledge that one of the limitations of this study was the lack of a control group, which we mention in our Discussion on page 10. We have now added additional text for clarity.

The major limitations of this study were that mental wellbeing was assessed using only one outcome measure (SWEMWBS) and that the intervention took place in a complex environment with no control group. However, to account for the absence of a control group, a 27% deadweight percentage was applied. This percentage is derived from rigorous wellbeing valuation methodologies [43].

We have also added a reference for the 27% deadweight percentage:

43. Dancer, S. (2014). Additionality Guide: Fourth Edition 2014. Home and Communities Agency. Retrieved from

https://www.gov.uk/government/uploads/system/uploads/attachment_data/file/378177/ad

Page 10

Page 13

·      Please, to facilitate readiness write down the formula of SROI and explain each component in order. 

We have added text at the start of the Methods section to introduce the six stages of SROI.

The SROI evaluation process, as outlined in the 2012 Cabinet Office Guide [33], typically involves six key stages: 1) identifying stakeholders, 2) developing a theory of change model, 3) calculating inputs, 4) evidencing and valuing outcomes, 5) establishing impact, and 6) calculating the SROI ratio. Each of these stages is discussed below.

Page 3

Methods

·      Please explain in more detail what is collected in the SWEMWBS. Which items do you use and the scales, mostly for the outcomes used?

We have added a paragraph in the Methods section stating:

To calculate social value using the SWEMWBS, participants complete the questionnaire at baseline and follow-up time points. Total scores are calculated for both time points. Each total score is assigned a monetary value, and the change in value between baseline and follow-up is determined. Finally, 27% is subtracted from this change to account for deadweight (what would have happened anyway), resulting in the final social value for each participant.

Page 4

·      The authors must explain how the social value estimates were calculated. Please include the methodology in the paper. 

We have added text (and a table) in the Methods section to provide more detail of what monetary values were used from wellbeing valuation based on the SWEMWBS scores.

The monetary values below (see Table 1) indicate the amount of money the average person would need to receive in order to experience an equivalent increase in their overall SWEMWBS score [36].

Table 1. SWEMWBS scores and monetary values based on wellbeing valuation

Overall SWEMWBS score

Monetary value based on wellbeing valuation

7 - 14

£0

15 -16

£9,639

17 - 18

£12,255

19 - 20

£17,561

21 - 22

£21,049

23 - 24

£22,944

25 - 26

£24,225

27 - 28

£24,877

29 - 30

£25,480

31 - 32

£25,856

33- 34

£26,175

35

£26,793

Pages 4 and 5

·      It is necessary to include some characteristics of the sample evaluated besides gender. It would be important to know how bad their health condition was, what condition they have, what is their family situation, what were their financial status, and their working conditions. This is relevant to understand who is really benefiting from the service. It may be that those who are better off take over the service, and those in worse situations avoid the service as they do not have time. 

In addition to gender, age and ethnicity were collected from study participants. See text below:

With Perinatal referrals, BAME made up around half of all service users (with a ratio of 1:1.1), reflecting that many service users were referred through the Non-English Speaking Team (NEST) service. However, the ratio of BAME participants was lower in the baseline SWEMWBS and SROI respondents (at 1:0.7 and 1:0.8, respectively) indicating that BAME perinatal service users were under represented in the SROI evaluation of Perinatal services.

The average service user age was 51.7 (±17.0), with the age of Primary Care referrals (52.4±17.1), being marginally higher than Secondary Care referrals (47.9±10.1). Perinatal referrals were younger (32.3±6.7), as expected for this cohort. The age of baseline SWEMWBS respondents and SROI respondents was generally in line with total service users.

Unfortunately, data on participants’ health condition, family situation, financial status and working conditions were not collected.

Page 6

Discussion

·      Could you please clarify why you consider this a natural experiment? People are referred to the CAP (treatment), but they can decide whether to get it or not. Also, referrals are made to people under certain circumstances. A natural experiment would randomly assign people to CAP without any anticipation, so it is possible to observe changes over time.

We have added text to explain why we considered the CAP intervention a natural experiment:

This study employed a natural experiment approach. This means the CAP intervention investigated was not deliberately designed or controlled by researchers. Instead, it occurred naturally within a complex real-world setting.

Page 10

Reviewer 2 Report

Comments and Suggestions for Authors

This manuscript is relevant and timely.  I would like to see more details please of what values were used from HACT based on the mental health score being mapped onto this.  This would be helpful to the reader. 

Author Response

Please see attached Word document

Reviewer’s comments

Response and action

Page number

Reviewer 2

This manuscript is relevant and timely.  I would like to see more details please of what values were used from HACT based on the mental health score being mapped onto this.  This would be helpful to the reader. 

We have added text and Table 1 in the Methods section to provide more detail of what values were used from HACT based on the SWEMWBS scores.

The monetary values below (see Table 1) indicate the amount of money the average person would need to receive in order to experience an equivalent increase in their overall SWEMWBS score [36].

Table 1. SWEMWBS scores and monetary values based on wellbeing valuation

Overall SWEMWBS score

Monetary value based on wellbeing valuation

7 - 14

£0

15 -16

£9,639

17 - 18

£12,255

19 - 20

£17,561

21 - 22

£21,049

23 - 24

£22,944

25 - 26

£24,225

27 - 28

£24,877

29 - 30

£25,480

31 - 32

£25,856

33- 34

£26,175

35

£26,793

Pages 4 and 5

Reviewer 3 Report

Comments and Suggestions for Authors

  I think there are two points that the authors do not emphasize enough

1/ "attribution". As I understand it, the authors solve the issue of attribution (in a design where there is no control group) by asking 16 respondents to self-assess the attribution (qualitatively). This should be emphasized in the abstract. It should also be discussed in the section on strengths and weaknesses. In some way, this is original; and it could be emphasized. But this could be seen risky, and biased; then it should be recognized and discussed. (e.g. There are no "confidence intervals" on that).

2/ the only outcome measured is SWEMWBS scores ; this is a big limitation to restrict the effect of the program (and its welfare valuation) on only one mental health indicator. Here again, possible weakenesses should be recognized and discussed.

In short, the fragility of the results should be more emphasized. Maybe the program is even more efficient ! maybe less.. There are no confidence intervals (discussion of !!)

Author Response

Please see attached Word document

Reviewer’s comments

Response and action

Page number

Reviewer 3

  I think there are two points that the authors do not emphasize enough

1/ "attribution". As I understand it, the authors solve the issue of attribution (in a design where there is no control group) by asking 16 respondents to self-assess the attribution (qualitatively). This should be emphasized in the abstract. It should also be discussed in the section on strengths and weaknesses. In some way, this is original; and it could be emphasized. But this could be seen risky, and biased; then it should be recognized and discussed. (e.g. There are no "confidence intervals" on that).

The HACT Mental Health Social Value Bank (MHSVB) utilises a wellbeing valuation framework that accounts for a 27% deadweight percentage when evaluating health interventions conducted without a control group. This adjustment is included in the MHSVB methodology and is supported by relevant research. We have added additional text below:

However, to account for the absence of a control group, a 27% deadweight percentage was applied. This percentage is derived from rigorous wellbeing valuation methodologies [43].

We have also added a reference for the 27% deadweight percentage:

43. Dancer, S. (2014). Additionality Guide: Fourth Edition 2014. Home and Communities Agency. Retrieved from

https://www.gov.uk/government/uploads/system/uploads/attachment_data/file/378177/additionality_guide_2014_full.pdf

For sensitivity analysis, attribution and displacement percentages were included to provide a more conservative estimate of social value. However, attribution and displacement are not mandatory inputs when using the MHSVB. Nevertheless, we have added text to clarify that when attribution is considered, the overall range of SROI ratios is £1 invested for every £3.40 to £4.69 of social value created:

Given the limited sample size of interviewees providing attribution data (n=14), a sensitivity analysis was conducted utilising the reported attribution levels (72.5% for Primary Care and Secondary Care, 60% for Perinatal) to establish a potential range of SROI values. Since the Primary Care and Secondary Care pathways represent over 90% of referrals, the attribution level of 72.5% was used for the sensitivity analysis. This resulted in an overall SROI range of £1 : £3.40 -£4.69.

Page 10

Page 13

Page 8

2/ the only outcome measured is SWEMWBS scores; this is a big limitation to restrict the effect of the program (and its welfare valuation) on only one mental health indicator. Here again, possible weaknesses should be recognized and discussed.

We have now stated the major limitations of this study were the use of only one outcome measure (SWEMWBS) and the absence of a control group.

However, the major limitations of this study were that mental wellbeing was assessed using only one outcome measure (SWEMWBS) and that the intervention took place in a complex environment with no control group. However, to account for the absence of a control group, a 27% deadweight percentage was applied. This percentage is derived from rigorous wellbeing valuation methodologies [43].

Page 10

In short, the fragility of the results should be more emphasized. Maybe the program is even more efficient! maybe less.. There are no confidence intervals (discussion of !!)

We have changed the text in the conclusion to convey the potential impact of CAP while acknowledging the limitations of the available evidence.

The findings suggest that CAP, a welfare and social prescribing support service, may demonstrate effectiveness in generating positive social value by improving self-reported wellbeing among individuals experiencing or at risk of financial or social hardship.

Pages 10 and 11
